# Agroecological Risk Assessment Based on Coupling of Water and Land Resources—A Case of Heihe River Basin

**Jiashan Yu [1], Jun Zhou [2], Jing Zhao [1,\*], Ran Chen [1], Xueqi Yao [1], Xiaomin Luo [1], Sijia Jiang [1] and Ziyang Wang [1]**

[1]    School of Landscape Architecture, Beijing Forestry University, Beijing 100083, China
[2]    Faculty of Architecture, Civil and Transportation Engineering, Beijing University of Technology, Beijing 100124, China
\*    Correspondence: zhaojing@bjfu.edu.cn

**Abstract:** In the arid zone of northwest China, the Heihe River Basin (HRB), as a typical inland river basin, has a fragile regional ecological environment, obvious ecological degradation characteristics, and extremely serious problems in the utilization of agricultural land resources. Meanwhile, the shortage of water resources, the low reduction of land quality, and excessive agricultural activities have greatly increased the local water and land pressure. In this paper, firstly, using the Malmquist DEA model and coupling coordination degree model, the agroecological risk assessment system on account of the coupling of water and land resources (WLR) is constructed. Secondly, taking HRB from 1995 to 2020 as an example, we carry out spatial correlation analysis based on the degree of risk-correlated WLR. Thirdly, we analyze the evolution process and spatial correlation of ecological risk of agricultural WLR in the HRB at the county scale, then we conclude and put forward policy suggestions for improvement. The results show that: (1) On the whole, the average ecological risk of agricultural water resources in the HRB from 1995 to 2020 was 0.933, indicating that the risk was declining; the average ecological risk of agricultural land resources in the HRB from 1995 to 2020 was 0.938, indicating that the risk was declining also. (2) The degree of ecological risk coupling and coordination of agricultural soil and water resources upstream of the HRB is on the rise, while that in the middle and lower reaches is on the decline. (3) Through panel model analysis, the matching suitability of WLR drives agroecological risk. The correlation between them is positive. In conclusion, this method can effectively evaluate the agroecological risk of WLR and provide technical support for agricultural production and management in arid areas.

**Keywords:** agricultural WLR; ecological risk; data envelopment analysis; temporal and spatial variation; HRB



## 1. Introduction

Arid and semi-arid areas of China account for 52.5% of its total territory. Affected by climate change and human activities, China is one of the most drought-prone countries globally. The area of the arid zone is about 2.8 million square kilometers, stretching from the northwest border to the west and reaching the west foot of the Great Khingan Mountains in the east. It includes about 965 counties in 16 provinces, cities, and autonomous regions, and its total area accounts for more than half of the national area. However, Northwest China covers 83% of China's arid and semi-arid regions. At present, with the acceleration of the urbanization process, a series of activities such as over-cultivation and overgrazing by humans have caused an increase in the pressure on agricultural WLR, further increasing the agroecological risks and thus adversely affecting the ecosystem and sustainable development of society. So, the research on ecological risk assessment of agricultural WLR in arid areas is a problem worth paying attention to.

Ecological risk assessment generally refers to the assessment of the risk to an ecosystem or its components. Most of the existing ecological risk studies have focused on metal

pollution sources and landscapes [1], and ecological risk assessment specifically for agriculture is very rare [2]. In the ecological risk assessment for agriculture, current scholars' studies on the ecological risk of soil and water resource utilization usually start from water or land resources alone, and we are short on the perspective of water and land resource coupling. For example, Jiang et al. [3] used the average Dee's decomposition method to assess the risk of water scarcity in Heilongjiang province and its 13 prefecture-level cities based on the entropy-weighted physical element model. Yang [4] constructed a coupled principal component analysis and data envelope model for land ecological risk assessment based on defining characteristics and connotations for the study of ecological risks of land use in the Changzhutan urban agglomeration. In addition, studies on agricultural WLR are mostly from the perspective of water and land matching. For example, Nan [5] applied the WLR matching index to measure the matching status of agricultural WLR in the northwest dry zone and estimated the potential of agricultural WLR utilization under two scenarios. The lack of a coupled water and land risk perspective in current relevant studies has led to the inability of subsequent researchers to collaboratively optimize the sustainable use of water and land in drylands. Many measures of ecological risk are landscape-level, and few studies consider water and land use efficiency in a comprehensive way. In terms of the ecological risk research method, it is mostly from the perspective of landscape ecology. For example, Zhang et al. [6] took the Shiyang River basin, a typical basin of arid inland rivers, as the research object and evaluated the spatial and temporal changes of ecological risk in the basin based on the landscape pattern. Studies on landscape patterns ignore the influence of socioeconomic factors, and their results are difficult to use to guide urban planning. In addition, there are fewer studies on the ecological risk of coupled agricultural WLR and even fewer comprehensive studies involving their matching relationship with WLR. So it is very necessary to propose a measurement method for water and land ecological risk from the perspective of efficiency.

Therefore, this study first focuses on the spatial and temporal changes in ecological risk of agricultural WLR and uses the (1) Malmquist DEA model and the coupling index to construct a more complete agroecological risk assessment system for coupled WLR, assessing the ecological risk of agricultural WLR use in the HRB from 1995 to 2020. (2) The panel model is used to explore the spatial driving effects of water and land ecological risks and propose policy improvement recommendations. This study complements the research method of ecological risk and provides references for the rational exploitation and management of agricultural WLR in arid areas.

## 2. Study Area

In this study, we focused on the risk of agricultural WLR in the HRB in China. The HRB is a typical inland river basin with a fragile regional ecological environment, obvious ecological degradation characteristics, and extremely serious agricultural land resource utilization issues. Since the Western Region Development of China in 2000, the HRB has strengthened comprehensive management to restore the downstream ecological environment, and a water diversion policy was implemented in 2001 [7]. In August 2001, the State Council approved the State Letter (2001) No. 86 "The Recent Management Plan of the HRB", suggesting a 3-year management plan to realize the "Water Allocation Plan for the Main Stream of the Black River", approved by the State Council (Water Administration (1997) No. 496), while forming a comprehensive management and protection system for the ecosystem, focusing on the rational allocation of water resources [2]. In 2015, the Ministry of Agriculture and other departments issued "Promoting Sustainable Development of Agriculture and Animal Husbandry in the Northwest Dry Zone", which proposes to protect land and water resources, strengthen ecological protection, enhance ecological service functions, and take other views. In September 2018, the Ministry of Water Resources issued "Management Measures for Water Scheduling in the Main River of the Black River" to strengthen the unified scheduling of water in the main river of the Black River and

reasonably allocate water resources in the HRB to promote ecological improvement and social-economic development in the area.

Being the second largest inland river basin in China, the HRB includes over 30 tributaries with total average multi-year water resources of 4.173 billion m$^3$ [8]. With a basin area of about 142,900 km$^2$, the watershed is divided into upper, middle, and lower parts by Yingluo Gorge and Zhengyi Gorge (Figure 1), and their water resource conditions are very different. The part above Yingluo Gorge is the upper part, which is the main flow-producing area and includes a part of Qilian County, Haibei Prefecture, and Qinghai Province. The part from Yingluo Gorge to Zhengyi Gorge, the main water use area, is the middle reaches and includes six districts and counties in the Zhangye area of Gansu Province and two districts under the jurisdiction of the Jiuquan area. The part below the Zhengyi Gorge is downstream and belongs to the extremely arid zone [9], including a part of Jinta County in the Jiuquan region of Gansu and the Ejina Banner of Inner Mongolia Autonomous Region. With the accelerated population expansion and the increase of economic activities in the basin, the HRB is experiencing a serious shortage of water resources, an increase in the area of arable land, but a decrease in the area of grassland, which leads to ecological problems such as desertification of the land. Restoration of the ecological environment of the basin has become a prerequisite and the basis for sustainable development of the area [10].

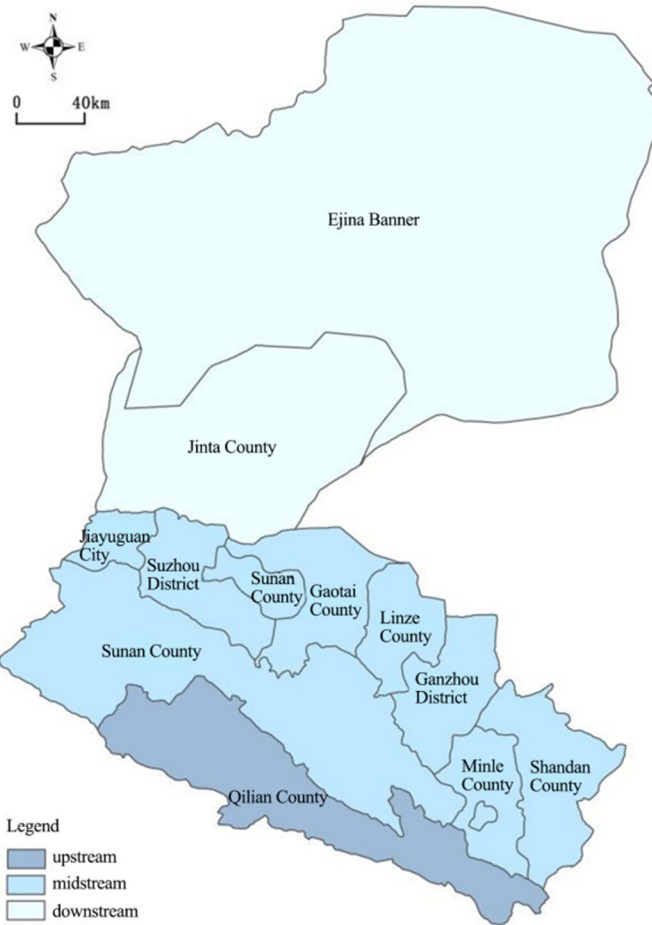

**Figure 1.** Division of districts and counties in HRB.

## 3. Data and Method

### 3.1. Data Sources

In this study, data related to various types of land areas, agricultural water consumption, water pollution level, and irrigated area in the HRB were used. Data on various land areas were derived from the land use data of the HRB in 1995, 2000, 2005, 2010, 2015, and 2020. Water resources data were obtained from the Water Resources Bulletin of Qinghai

Province, the Water Resources Bulletin of Inner Mongolia Autonomous Region, the Water Resources Bulletin of Gansu Province, the Ecological and Environmental Status Bulletin, the China Water Resources Yearbook, the China Water Resources Bulletin, the China Water Resources Statistical Yearbook (1995–2020), and the National Tibetan Plateau Scientific Data Center [11–13]. Socioeconomic data, such as irrigated area, were obtained from the Qinghai Statistical Yearbook, Qinghai Province Environmental Statistics Bulletin, Inner Mongolia Statistical Yearbook, Gansu Development Yearbook, Gansu Yearbook, Zhangye Statistical Yearbook, Jiayuguan Statistical Yearbook, Urban Statistical Yearbook (1995–2020), and ecological and environmental department websites. The method of searching for the literature [14] was applied to obtain water consumption data for some missing years. The study area and graphic output in this study are based on ArcGIS software Version 10.6.

*3.2. Method*

3.2.1. Data Envelopment Analysis

Since agroecological risks are relative among districts and counties, and this study explores the time series changes of ecological risks in agricultural WLR, the Malmquist DEA model, a non-parametric method based on relative efficiency and capable of analyzing time series changes, was chosen.

Invoking Efficiency Assessment Methods for Risk Assessment

Data envelopment analysis (DEA), first proposed by Charles Cooper and Rhodes in 1978, is a nonparametric method based on relative efficiency for evaluating the input-output efficiency of decision units [15]. Its distinctive feature is that it does not require parameters that do not require pre-estimation as well as hypothetical weights. In this study, the input-output model of efficiency assessment, i.e., the DEA method, is invoked to assess the ecological risk of agricultural WLR use in the HRB. The formula is as follows:

$$R_L = \frac{\sum_{j=1}^{n} u_i Output_{jr}}{\sum_{i=1}^{n} v_i Input_{ir}} \tag{1}$$

The ecological risk of agricultural WLR use is defined as the ratio of the input agroecological risk factor to the output agricultural WLR use factor [4] In the equation, $R_L$ is the ecological risk value of agricultural WLR use, and output and input are the agroecological risk factor and agricultural WLR use factor, respectively [16].

DEA Model Malmquist Index

The Malmquist productivity index was proposed by Malmquist (1953) based on the DEA model, which uses the ratio of distance functions to calculate input-output efficiency, thus objectively measuring the relationship between technical efficiency changes, technological changes, and total factor changes [17] and enabling accurate time series change analysis [18]. In this paper, the Malmquist index is understood as the ecological risk arising from agricultural WLR use, and the land use data of the HRB for 1995, 2000, 2005, 2010, 2015, and 2020 are selected as the basic information source, combined with socioeconomic data and other data to dynamically assess the ecological risk of agricultural WLR in the study area, respectively. It is defined as follows [19]:

$$M = (M_t \cdot M_{t+1})^{\frac{1}{2}} = \left[ \frac{L_0^t(x^{t+1}, y^{t+1})}{L_0^t(x^t, y^t)} \times \frac{L_0^{t+1}(x^{t+1}, y^{t+1})}{L_0^{t+1}(x^t, y^t)} \right]^{\frac{1}{2}} \tag{2}$$

$$M_t = Effch \times Tech$$
$$Effch = \frac{L_0^t(x^{t+1}, y^{t+1})}{L_0^t(x^t, y^t)}$$
$$Tech = \left[ \frac{L_0^t(x^{t+1}, y^{t+1})}{L_0^t(x^{t+1}, y^{t+1})} \frac{L_0^t(x^t, y^t)}{L_0^t(x^t, y^t)} \right]^{\frac{1}{2}} \tag{3}$$

$$M_t = \text{Pech} \times \text{Sech} \times \text{Tech}$$
$$\text{Pech} = \frac{L_v^t\left(x^{t+1}, y^{t+1}\right)}{L_v^t\left(x^t, y^t\right)}$$
$$\text{Sech} = \frac{L_0^t\left(x^{t+1}, y^{t+1}\right)}{L_v^t\left(x^{t+1}, y^{t+1}\right)} \Big/ \frac{L_0^t\left(x^t, y^t\right)}{L_v^t\left(x^t, y^t\right)} \tag{4}$$

In Equation (2), M is the Malmquist productivity index, which is interpreted in this paper as the ecological risk due to agricultural WLR use, and $M_t$ and $M_{t+1}$ are the Malmquist productivity indices for periods t and t+1, respectively. In the time period from t to t+1, a larger value of M indicates a greater ecological risk in that time period, while M > 1 implies an elevated risk in that time period. In the case of constant returns to scale, M can be decomposed into the technical efficiency index (Effch) and Tech indices, which denote the technical efficiency index and the technical progress index, respectively, and the Effch in the case of variable returns to scale is further decomposed into the pure technical efficiency index (Pech) and the scale efficiency index (Sech) [20]. The above four decomposition indices help in the interpretation of the results.

In view of the advantages and disadvantages of DEA, the selection of input-output indicators should avoid linear correlation among indicators as much as possible, while considering the availability of data and referring to the relevant literature to select evaluation indicators [20–24]. For the ecological risk of agricultural water resources, the total water consumption of agriculture, forestry, livestock, and fisheries (million $m^3$), the total water consumption of agricultural life (million $m^3$), the water area ($km^2$), and the water area of agriculture, forestry, livestock, and fisheries ($km^2$) are input indicators $x_1$, $x_2$, $x_3$ and $x_4$, and the degree of water quality pollution is the output indicator $y_1$. For the ecological risk of agricultural land resources, the area of farmland ($km^2$), the area of unused land ($km^2$), and the vegetation cover index are input indicators $x_1$, $x_2$ and $x_3$, and the rate of land degradation and the landscape fragmentation index are output indicators $y_1$ and $y_2$, as shown in Table 1.

**Table 1.** Input-output indicators for agroecological risks of WLR use.

| | Water Resources Ecological Risk | Land Resources Ecological Risk |
| --- | --- | --- |
| Input indicators | Total water consumption of agriculture, forestry, livestock, and fisheries (million $m^3$) | Area of farmland ($km^2$) |
| | Total water consumption of agricultural life (million $m^3$) water area ($km^2$) | Area of unused land ($km^2$) Vegetation cover index |
| | Water area of agriculture, forestry, livestock, and fisheries ($km^2$) | |
| Output indicators | Degree of water quality pollution | Rate of land degradation Landscape fragmentation index |

### 3.2.2. Water and Land Risk Coupling Index

Coupling refers to the interaction of two or more systems, a concept originally used in the physics of electronics and subsequently extended by many scholars to social development systems with the following formula:

$$C = 2\sqrt{\frac{R_w \cdot R_s}{(R_w + R_s)^{\frac{1}{2}}}} \tag{5}$$

$$T = a_w \cdot R_w + a_s \cdot R_s \tag{6}$$

$$D = \sqrt{C \times T} \tag{7}$$

In Equation (5) C is the coordination index, $R_w$ and $R_s$ is the combined value of agricultural water resources ecological risk and land ecological risk, respectively; in Equation (6) T is the comprehensive evaluation index, where it is assumed that the contribution of water resources and land resources to agroecological risk is the same, both $a_w$ and $a_s$ are set to 0.5; in Equation (7) D is the coupling index. Drawing on the principle that the coupled agroecological risk coordination index is used to evaluate the risk of the transportation system in the literature [25], this study measured the coupling index between the ecological risk of agricultural water resources and the ecological risk of agricultural land resources to explore the degree of agroecological risk of WLR coupling. The degree of coordination of the ecological risk of agricultural WLR in the HRB was divided into five levels according to the size of the coupling index, as shown in Table 2.

**Table 2.** Coupled agroecological risk coordination index of WLR in the HRB.

| Coordination Type | Basic Disorder | Basic Coordination | Well Coordination | Extremely Well Coordinated | Over-Coordination |
|---|---|---|---|---|---|
| Index of coupling | 0–0.5 | 0.5–0.8 | 0.8–1.0 | 1.0–1.2 | >1.2 |

### 3.2.3. WLR Matching Factor

The WLR matching coefficient (R) is the spatial and temporal relationship between the amount of WLR available for agricultural production in a region. The higher the degree of agricultural WLR matching, the better the basic conditions for agricultural production [26]. It is defined as follows:

$$R_i = \frac{W_i \cdot a_i}{L_i}(i = 1, 2, 3 \ldots\ldots n) \tag{8}$$

where $R_i$ is the matching coefficient of agricultural WLR in the first area of the HRB (104 $m^3$/$hm^2$); $W_i$ is the volume of water resources available in the i area (108 $m^3$); $a_i$ is the proportion of water consumption in agriculture, forestry, livestock, and fisheries to the total water consumption in the i area; $L_i$ is the area of water consumption in agriculture, forestry, livestock, and fisheries in the i area (104 $hm^2$); n is the number of areas within the administrative division of the HRB n = 11; the time series of this study is 6.

### 3.2.4. Panel Model

The changes in WLR matching coefficients and the ecological risk changes in agricultural WLR in various districts and counties in HRB were regressed through panel models. In panel data analysis, it is typical to assume that each object has unobservable fixed characteristics that affect the dependent variable (this is the so-called unobserved heterogeneity of objects). In this case, the panel data model is

$$y_{it} = x_{it}\beta + c_i + v_{it} \tag{9}$$

where $c_i$ is unobservable and called an individual effect, $\beta$ is fixed parameters. If $c_i$ and $x_{it}$ are uncorrelated, the effect is referred to as a random effect: $c_i$ and $x_{it}$ can be correlated, the effect is referred to as a fixed effect. In this study, the coupling index is taken as the dependent variable. The coupling index, population density, GDP density, and urbanization rate are combined as independent variables to explore the relationship between the matching coefficient of WLR and agricultural ecological risk changes.

## 4. Results

### 4.1. Ecological Risk Assessment of Land and Water Resources in the HRB

The DEAP 2.1 model was used to measure the Malmquist index of 11 districts and counties in the HRB from 1995 to 2020 to obtain the ecological risk values of the WLR by time period and sub-region.

4.1.1. Ecological Risk Assessment of Agricultural Water Resources in HRB

As a whole, the mean value of the ecological risk of agricultural water resources in the HRB from 1995 to 2020 is 0.933, indicating a decreasing trend in risk (Table 3). The mean value of the technological change index (Techch) is 0.872, indicating technological progress in reducing the ecological risk of agricultural water resources. The ecological risk of water resources changed from greater than 1 to less than 1 in 2005, indicating that the ecological risk changed from an increase to a decrease in 2005. In 2000, after the diversion of the Heihe River, the dynamic balance of the water cycle in the midstream and downstream was broken, which caused an ecological risk to water resources to some extent. By 2005, the effectiveness of water diversions was evident, resulting in a decrease in ecological risk to water resources [27].

**Table 3.** Agroecological risk index and decomposition of water in the HRB by time period, 1995–2020.

| Time Period | Risk Index (Tfpch) | | | | |
| | Technical Efficiency Index (Effch) | | | Technological Progress Index (Techch) | |
| | Pure Technology Efficiency Index (Pech) | Scale Efficiency Index (Sech) | Technical Efficiency Changes (Effch) | Technological Progress (Techch) | TFP Efficiency (Tfpch) |
|---|---|---|---|---|---|
| 1995–2000 | 1.243 | 1.069 | 1.329 | 0.793 | 1.053 |
| 2000–2005 | 0.991 | 1.114 | 1.104 | 1.163 | 1.284 |
| 2005–2010 | 0.943 | 1.027 | 0.969 | 0.861 | 0.834 |
| 2010–2015 | 1.076 | 0.987 | 1.062 | 0.716 | 0.761 |
| 2015–2020 | 0.893 | 1.035 | 0.924 | 0.889 | 0.822 |
| Mean value | 1.022 | 1.046 | 1.069 | 0.872 | 0.933 |

In terms of the Effch, the index is greater than 1 for 1995–2005 and 2010–2015, with the maximum being 1.329 for 1995–2000. This indicates that the technical efficiency in the above time period is not sufficient to reduce the ecological risk of agricultural water resources. In terms of Techch, only the index for 2000–2005 is greater than 1, at 1.027, which indicates that the technological progress in this time period is insufficient. From the Pech, it is fluctuating, and the index is less than 1 for 2000–2010 and 2015–2020, which indicates that the overall management of agricultural water resources in that time was strong. From the Sech, the scale efficiency of agricultural water resource use becomes smaller in 2010–2015, thus reducing the ecological risk. Since 2000, with the accelerated population growth, the arable land area has expanded year by year, leading to an increase in agricultural water demand [28], thus increasing its ecological risk. At the same time, the level of technology after 2005 is sufficient to reduce the ecological risk of water resources.

From the spatial level, the ecological risk of agricultural water resources in the HRB from 1995 to 2020 is midstream > downstream > upstream, and the comparative results of each district and county are: Linze County > Shandan County > Gaotai County > Ganzhou District > Qilian County > Minle County > Sunan County > Jiayuguan City > Jinta County > Suzhou District > Ejinabari (Table 4). Among them, the risks in Linze County, Shandan County, Gaotai County, and Ganzhou District are greater than 1, and the Linze County risk is the largest at 1.144. The remaining seven districts and counties' risk is lower; the lowest Ejinan Banner is 0.645.

**Table 4.** Agroecological risk of water index and decomposition by region in the HRB, 1995–2020.

| Region | Risk Index (Tfpch) | | | | |
|---|---|---|---|---|---|
| | Technical Efficiency Index (Effch) | | | Technological Progress Index (Techch) | |
| | Pure Technology Efficiency Index (Pech) | Scale Efficiency Index (Sech) | Technical Efficiency Changes (Effch) | Technological Progress (Techch) | TFP Efficiency (Tfpch) |
| Ejina Banner | 0.844 | 0.945 | 0.798 | 0.809 | 0.645 |
| Jinta County | 0.963 | 0.991 | 0.954 | 0.879 | 0.839 |
| Suzhou District | 0.959 | 0.994 | 0.953 | 0.853 | 0.813 |
| Jiayuguan City | 1.000 | 1.000 | 1.000 | 0.874 | 0.874 |
| Gaotai County | 1.047 | 1.228 | 1.285 | 0.850 | 1.093 |
| Sunan County | 1.000 | 1.000 | 1.000 | 0.911 | 0.911 |
| Linze County | 1.051 | 1.239 | 1.302 | 0.878 | 1.144 |
| Ganzhou District | 1.310 | 0.892 | 1.169 | 0.862 | 1.007 |
| Shandan County | 1.072 | 1.214 | 1.302 | 0.853 | 1.110 |
| Minle County | 1.056 | 1.065 | 1.125 | 0.858 | 0.965 |
| Qilian County | 1.000 | 1.000 | 1.000 | 0.980 | 0.980 |
| Mean value | 1.022 | 1.046 | 1.069 | 0.872 | 0.933 |

From Figure 2, we can know that the ecological risk of upstream agricultural water resources is decreasing most of the time, with an increasing trend from 1995 to 2000; the midstream risk is more variable, with a decreasing trend from 2010 to 2015; and the ecological risk of downstream agricultural water resources is increasing most of the time, with a decreasing trend from 1995 to 2000 and from 2005 to 2010.

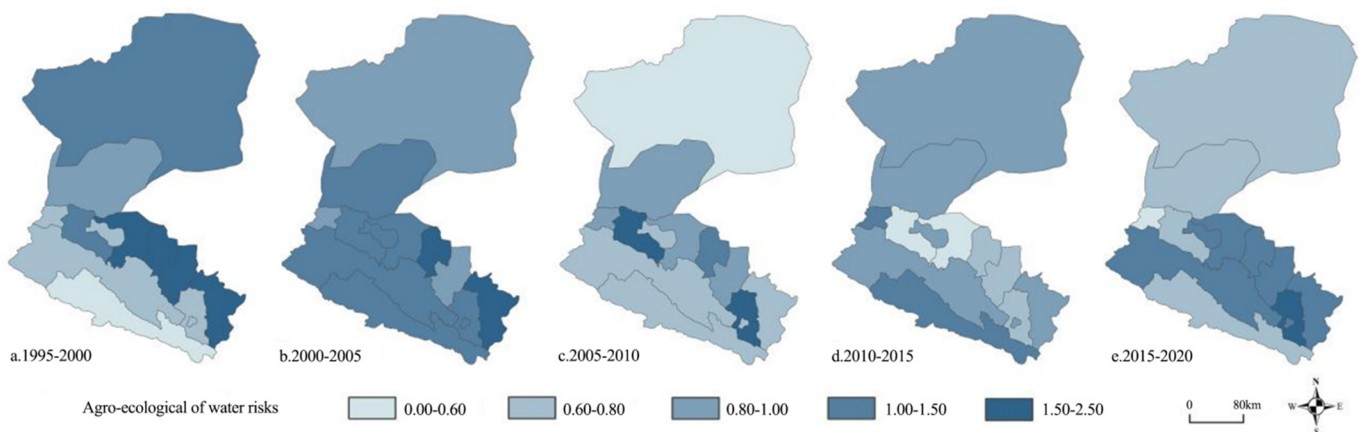

**Figure 2.** Water risk metric agroecological risk of land by district and county in the HRB over time.

4.1.2. Ecological Risk Assessment of Agricultural Land Resources in the HRB

As a whole, the mean value of the ecological risk of agricultural land resources in the HRB for 1995–2020 was 0.938 (Table 5), indicating a decreasing trend in risk. The mean values of Techch and Pech were less than 1, indicating that the progress of Techch and the management level reduced the ecological risk of agricultural land resources. The values of ecological risk of land resources for 2005–2010 and 2015–2020 were greater than 1, indicating a decreasing-increasing-decreasing-increasing trend of risk.

**Table 5.** Agroecological risk index and decomposition of land in the HRB by time period, 1995–2020.

| Time Period | Risk Index (Tfpch) | | | | |
| --- | --- | --- | --- | --- | --- |
| | Technical Efficiency Index (Effch) | | | Technological Progress Index (Techch) | |
| | **Pure Technology Efficiency Index (Pech)** | **Scale Efficiency Index (Sech)** | **Technical Efficiency Changes (Effch)** | **Technological Progress (Techch)** | **TFP Efficiency (Tfpch)** |
| 1995–2000 | 0.998 | 1.014 | 1.012 | 0.971 | 0.983 |
| 2000–2005 | 1.028 | 1.018 | 1.046 | 0.812 | 0.849 |
| 2005–2010 | 0.972 | 1.049 | 1.020 | 1.230 | 1.255 |
| 2010–2015 | 1.005 | 0.990 | 0.995 | 0.647 | 0.644 |
| 2015–2020 | 0.985 | 0.961 | 0.946 | 1.138 | 1.077 |
| Mean value | 0.997 | 1.006 | 1.003 | 0.935 | 0.938 |

From the Effch, the index is greater than 1 for 1995–2010, where the risk reaches 1.046 for 2000–2005, indicating that the technical efficiency at that time was not sufficient to reduce the ecological risk of agricultural land resources. In terms of Techch, the index is greater than 1 for 2005–2010 and 2015–2020, with the index reaching 1.230 for 2005–2010, indicating insufficient technological progress at that time. From the Pech, agricultural land resources management was not strong enough in 2000–2015 and 2010–2015. From the Sech, the scale efficiency of agricultural land resources becomes less efficient in 2010–2020.

From the spatial level, the ecological risk of agricultural land resources in the HRB from 1995 to 2020 is midstream > upstream > downstream, and the comparative results of each district and county are: Jiayuguan City > Minle County > Shandan County > Suzhou District > Jinta County > Ganzhou District > Gaotai County > Sunan County > Linze County > Ejina Banner > Qilian County (Table S1). Among them, Jiayuguan City, Minle County, and Shandan County have risk values greater than 1. Jiayuguan City has the greatest risk of 1.038, and the other 8 districts and counties have a reduced risk.

From each decomposition index, the lack of technical level and management ability of the midstream districts and counties, except Suzhou District and Jiayuguan City, make their ecological risk of agricultural land resources higher.

Figure 3 shows that the ecological risk of upstream agricultural land resources decreases most of the time, with the trend increasing from 2000 to 2010; the ecological risk of midstream is less stable, and the risk in Jiayuguan City, Minle County, and Suzhou District increases more of the time; the downstream risk increased from 2000 to 2010, and the downstream risk in Jinta County increased from 2015 to 2020.

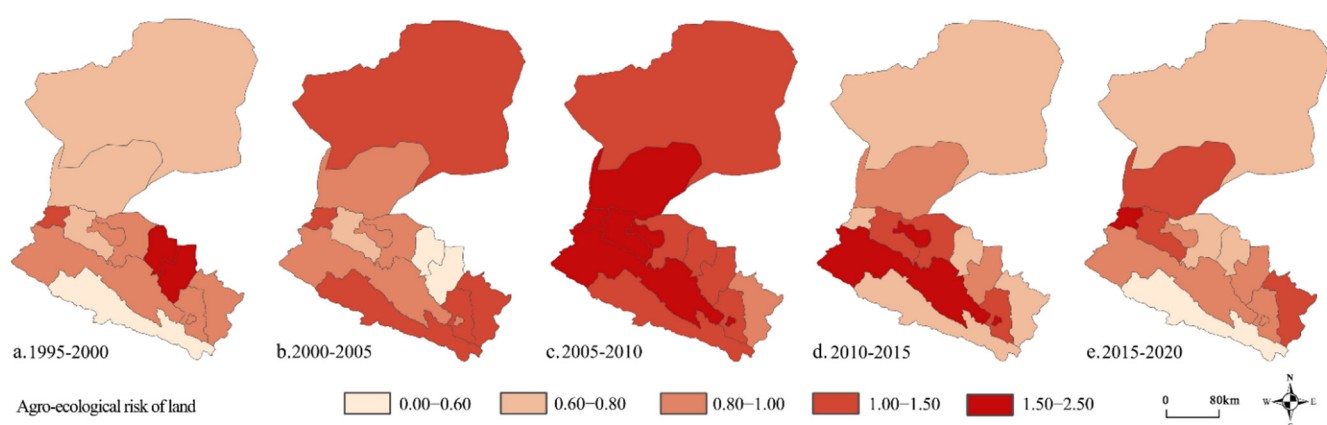

**Figure 3.** Agroecological risk of land by district and county in the HRB over time.

*4.2. Ecological Risk Coupling Relationship of Agricultural WLR*

In general, the greater the coupling index, the better the degree of coordination of the two attributes, i.e., the simultaneous increase in the values of the two attributes [29]. Since the two attributes for coupling analysis in this study are ecological risks of agricultural WLR, the better coordination of the two represents the simultaneous increase of the two risks, the situation is unfavorable to agroecological security and can be interpreted as a large integrated risk [25]. Therefore, the types of coupled coordination of WLR in Table 2 with a basic disorder, basic coordination, good coordination, extreme coordination, and excessive coordination are interpreted as low risk, lower risk, medium risk, higher risk, and high risk of agricultural WLR in that order. As can be seen from Figure 4, the degree of coupled coordination of ecological risk of agricultural WLR in the upper reaches of the HRB showed an increasing trend, rising 42.0% from 1995 to 2020; the degree of coupled coordination in the middle reaches and lower reaches showed a decreasing trend, decreasing 22.8% in the middle reaches and 12.0% in the lower reaches from 1995 to 2020. The upstream changed from low to high risk in 2000 and dropped to medium risk after 2005. Before 2000, the northwestern part of the midstream was at high risk, including Gaotai County, Linze County, Ganzhou District, and Shandan County; from 2000 to 2005, the overall risk of the midstream decreased, with only Shandan County remaining at high risk and Minle County rising to high risk; after 2005, most districts and counties in the midstream were at medium risk or below. Jiayuguan City and Minle County rose to high risk and higher risk, respectively, after 2015. In 2005, the downstream shifted from medium risk to higher risk and then to low risk after 2010, among which Jinta County's risk level has been at medium risk.

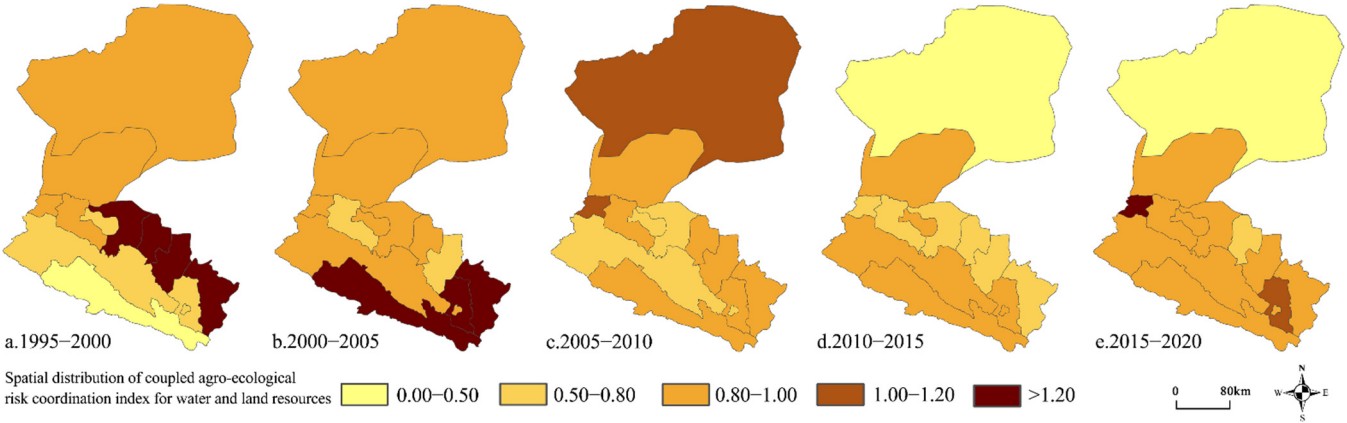

**Figure 4.** Spatial distribution of coupled agroecological risk coordination index for WLR in the HRB.

*4.3. Characteristics of Interannual Variation in Risk-Correlated WLR in the HRB*

Equation (8) was applied to calculate the matching coefficient of WLR in the HRB from 1995 to 2020, as shown in Table S2.

Spatially, the matching level of WLR in the HRB is upstream > midstream > downstream, and the comparison by districts and counties is: Jiayuguan City > Ganzhou District > Suzhou District > Linze County > Shandan County > Minle County > Qilian County > Jinta County > Gaotai County > Sunan County > Ejina Banner. Temporally, the matching level of WLR in the HRB from 1995 to 2020 showed a decreasing trend, and the differences among districts and counties gradually decreased. As can be seen from Figure 5, the matching level of WLR in the upper and middle reaches of the HRB was higher from 1995 to 2010; the matching level in the lower reaches did not change much from 1995 to 2020. The reasons for this result are mainly the large differences in the amount of agricultural WLR between districts and counties; the different climatic conditions in the upper, middle, and lower reaches; and the differences in the disaster situation. To some extent, it is also related to the differences in agricultural planting structure, regional economic level, and

urbanization level. In recent years, the inland river basin has developed high-efficiency water conservation demonstration areas and strengthened water conservation renovation of agricultural irrigation [30], thus saving agricultural water consumption, and the degree of risk-correlated WLR in the HRB has decreased as a result.

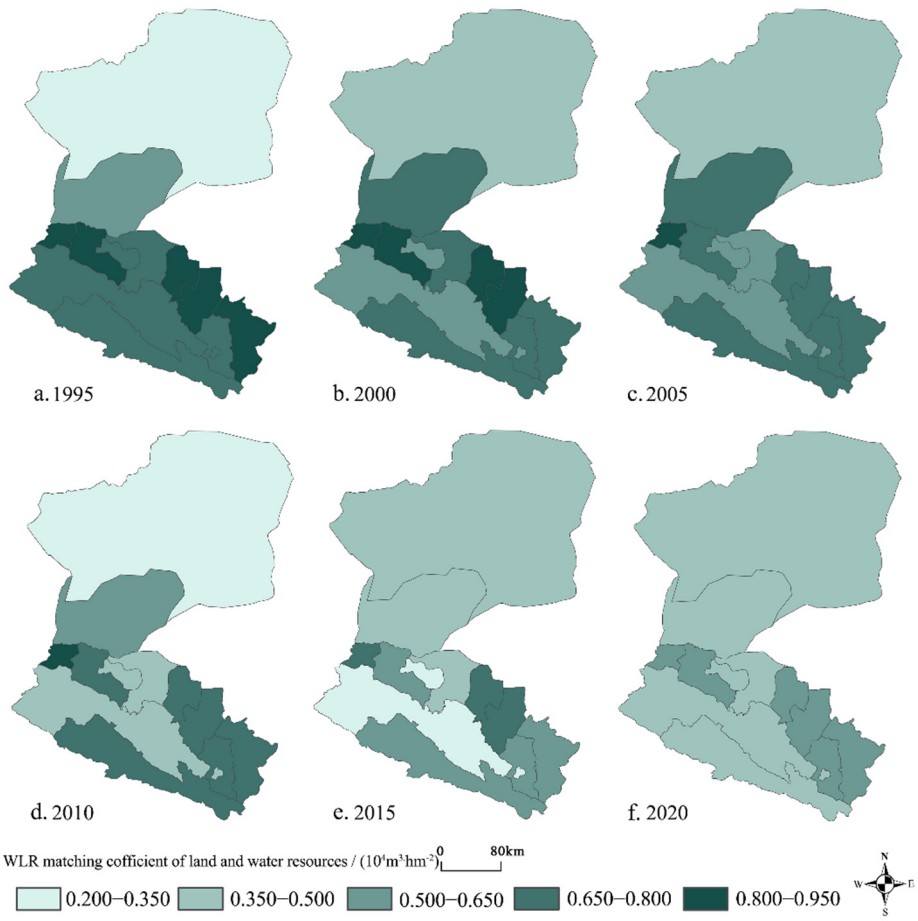

**Figure 5.** WLR matching coefficient of districts and counties in HRB by time.

After the western development strategy was launched in 2000, land resources were exploited on a large scale as arable land, and high water-consuming agriculture led to the intensification of problems such as over-extraction of groundwater resources and deterioration of land resources. At the beginning of policy implementation, the expansion of irrigated agricultural areas to promote economic development, coupled with an unreasonable planting structure, resulted in excessive irrigation water consumption, creating a situation in which the degree of risk-correlated WLR in the HRB has not improved since 2000 [31].

Qilian County in the upper part of the HRB is mainly mountainous, mainly developing forestry and animal husbandry, and has a more adequate water resources content, which is the reason why its WLR matching has been at a high level. However, due to its low overall development level and the burden of providing water resources and protecting ecological barriers in the middle and lower reaches, the degree of matching has gradually decreased in recent years. The topography of the middle reaches of the HRB is dominated by pre-mountain alluvial flood plains, wind-deposited plains, and low hills, which mainly develop the agricultural economy and are the most concentrated areas in the basin in terms of population and economy and the most intensive exploitation of water resources [32]. Among them, Jiayuguan City, Suzhou District, Ganzhou District, and Linze County have a relatively high degree of Risk-Correlated WLR, which is related to their higher level of economic development and modern agricultural technology. The topography of the

lower reaches of the HRB is dominated by Gobi, hills, and deserts, mostly unused land, with low precipitation and severe drought conditions. Ejina Banner is sparsely populated and mainly develops ecological civilization; Jinta County mainly develops agriculture and plantations; and the low degree of risk-correlated WLR after 2010 is mainly related to land degradation. The water distribution policy contributes more to reducing the difference in matching degrees among districts and counties.

*4.4. The Relationship between Water Ecological Risk and Matching Coefficient*

The regression analysis of the change of the WLR matching coefficient and the change of agricultural WLR ecological risk in each district and county of HRB showed that the change of the WLR matching coefficient was significantly correlated with the change of ecological risk of agricultural water resources utilization, and the coefficient of determination was 0.777 ($p < 0.05$), as shown in Table S3. This shows that the change in the WLR matching coefficient in HRB is positively correlated with the change in ecological risk of agricultural WLR utilization. After comparison, it is found that the effect is significant after adding the control variable of population density (POP), and the matching coefficient has a certain correlation with the population. With the increase of the matching coefficient of WLR, the ecological risk of water resources increases, while with the decrease of the matching coefficient of WLR, the ecological risk of resources decreases.

## 5. Discussion

### 5.1. Impact of Agricultural Policies on Water and Land Ecological Risks

In the agricultural water resources assessment, we find that from each decomposition index, the technical efficiency, technological progress, and management level of the downstream and midstream Suizhou districts of the HRB have achieved the purpose of reducing the ecological risk of water resources. The uneven distribution and unbalanced supply and demand of water resources in the HRB [28], as well as the implementation of the Heihe River water diversion policy, led to the rise of groundwater levels downstream and the serious decrease of water levels in the midstream, as well as the reduction of riparian vegetation in the basin, which further damaged the ecological environment [33]. The ecological risk of agricultural water resources in some districts and counties in the midstream was in an elevated trend. The water diversion measures implemented by the state since 2000 have gradually relieved the pressure on water resources in the downstream areas, reducing the ecological risk of agricultural water resources in the downstream year by year. The years 1995–2005 saw a high rate of population growth, and coupled with the implementation of environmental protection measures such as artificial oasis irrigation, the demand for water resources increased greatly and the ecological risk of water resources rose. By 2005, the effectiveness of water scheduling had gradually emerged, and coupled with advances in irrigation water-saving technology and the establishment of water-saving social structures in Ganzhou District and other districts and counties, the risk had been effectively reduced.

In terms of agricultural land resources, scale efficiency, and technology, these are the dominant factors limiting the ecological risk reduction of agricultural land use. With the urbanization of western arid regions, arable grasslands gradually tend to fragment [34], while factors such as water shortage and the natural climate also accelerate the desertification of agricultural land. With the deepening of human influence, the area of arable land in the midstream human activity-intensive area has increased significantly, and a large amount of desert and grassland has been reclaimed as arable land [26]. The inadequate supply of water resources, the decline in land quality, improper reclamation, overgrazing of grasslands, and irrational irrigation have caused phenomena such as soil salinization. Coupled with the gradual aridification of the climate, mobile sand dunes have increased the fragmentation of oasis outside the river banks and greatly reduced their area [31]. The land degradation index in the HRB from 2005 to 2010 was five times higher than in the previous time period, which directly led to an elevated ecological risk to the land. Since 2000,

the implementation of programs such as comprehensive management of grassland and return of farmland to forest and grass in the upper reaches of the HRB, the establishment of ecological projects of field protection forest networks in the middle reaches, and the construction of a protective forest system in the lower reaches initially increased the area of woodlands in the upper and middle reaches, and woodlands and grasslands in some areas showed degradation trends due to low rainfall [26]. Land degradation has improved by 2020.

*5.2. The Relationship between Water and Land Matching and Water and Land Risk*

The change in the WLR matching coefficient in HRB was positively correlated with the change in ecological risk of agricultural water resource utilization. Before 2000, the upstream was at low risk mainly due to low anthropogenic impact; the population concentration in the middle reaches and high risk in some districts and counties may be due to high population pressure, such as 6.4 times the average for Ganzhou District, 5.8 times for Linze County, 3.5 times for Gaotai County, and 2 times for Shandan County in China's arid zone [35], which put a great burden on the agroecosystem and subsequently led to land predatory management, the consequences of which are ecological problems such as land desertification and soil erosion; downstream, natural conditions are harsh and agricultural activities are scarce. The water diversion policy affects the supply and demand of water resources in the middle reaches, which in turn affects the upper reaches as a water source and increases the agroecological risk. From 2005 to 2010, the high risk in the lower reaches of Ejin Jinqi could be attributed to the deterioration of the surface water environment due to the implementation of a large number of projects required for water transfer [36]. After 2010, the adjusted water diversion policy and effective water management have reduced the agroecological risk in the lower reaches of Ejin Jinqi, and progress in agricultural water-saving irrigation technology and mechanization of production have put the upstream and midstream areas at medium-low risk. After 2015, the higher risk in the midstream Jiayuguan city may be due to pollution and desertification of its arable land due to urbanization [37], while the higher risk in the midstream Minle county may be due to the lagging construction of its government department management system and underdeveloped modern agricultural technology, etc. [38–48].

## 6. Conclusions and Recommendations

Based on the perspective of WLR coupling, this paper applies the input-output model and coupling model, combines the matching coefficient of WLR, analyzes the evolution process and spatial correlation of ecological risk of agricultural WLR in the HRB at the county scale, and mainly obtains the following conclusions.

The ecological risk of agricultural water resources in the HRB increased by 33.7% from 1995 to 2005 and decreased by 52.9% from 2005 to 2020, and in terms of spatial distribution, the ecological risk is midstream > downstream > upstream; the ecological risk of agricultural land resources decreased by 16.8% from 1995 to 2005, increased by 25.5% from 2005 to 2010, and decreased by 35.6% in 1995–2005, 25.5% from 2005 to 2010, 7.7% from 2015 to 2020, and in terms of spatial distribution, the ecological risk is midstream > upstream > downstream.

The degree of ecological risk coupling and coordination of agricultural WLR upstream of the HRB showed an increasing trend, with a total increase of 42.0% during the 25 years; the midstream and downstream showed a decreasing trend, with a total decrease of 22.8% in the midstream and 12.0% in the downstream during the 25 years.

The change in the matching coefficient of WLR in the HRB is positively related to the change in the ecological risk of agricultural WLR utilization, becoming the main driving factor. At the same time, it is found that the matching coefficient has a certain correlation with the population.

This study makes the following recommendations:

The middle reaches of the HRB should vigorously promote irrigation and water conservation technology, effectively utilize the recycled water cycle, strengthen the construction of modern agricultural facilities, increase mechanization input, scientifically adjust the agricultural planting structure, limit the arable land area to integrate oasis resources, support the livestock technology industry with policies to facilitate the efficient development of livestock farming, and strengthen environmental control in urban areas to prevent and control arable land pollution and desertification.

The upstream area of the HRB should improve the water retention intensity in the Qilian Mountains, use measures such as mountain closure and reforestation, protect water sources upstream, strengthen ecological environmental protection work, maintain the stability of the current oasis structure, strengthen the construction of livestock infrastructure, and focus on protecting grassland ecology.

Strengthen the sustainable use of agricultural WLR, macro management, and regulation of agricultural land resources and water resources; promote the balance of resource supply and demand; implement unified monitoring, management, and evaluation of WLR in the HRB; realize unified scheduling in the upstream, midstream, and downstream of the basin; and establish a green guarantee system. The downstream of the HRB should implement the construction project of the protective forest, the project of returning farmland to forest and grass, restore the vegetation cover in arid areas, and reasonably manage the implementation of water resources and hydraulic projects.

**Supplementary Materials:** The following supporting information can be downloaded at: https://www.mdpi.com/article/10.3390/land12040794/s1, Table S1: Agroecological risk of the land index and decomposition by region in the HRB, 1995–2020; Table S2: Matching coefficients agricultural of WLR in HRB from 1995 to 2020; Table S3: Regression equation between the matching coefficient of WLR and ecological risk of agricultural water resources.

**Author Contributions:** Conceptualization, J.Y., J.Z. (Jun Zhou) and J.Z. (Jing Zhao); methodology, X.Y. and X.L.; software, S.J.; validation, Z.W. and S.J.; formal analysis, J.Z. (Jun Zhou); investigation, X.Y.; resources, X.Y., J.Y. and X.L; data curation, Z.W.; writing—original draft preparation, S.J.; writing—review and editing, X.L., J.Z. (Jun Zhou), R.C. and X.Y.; visualization, S.J.; supervision, J.Z. (Jing Zhao); project administration, S.J.; funding acquisition, X.L. and J.Z. (Jing Zhao). All authors have read and agreed to the published version of the manuscript.

**Funding:** This work was supported in part by the National Natural Science Foundation of China under Grant 52208041, the Key Laboratory of Ecology and Energy-saving Study of Dense Habitat (Tongji University), Ministry of Education under Grant 20220110 and the Beijing High-Precision Discipline Project, Discipline of Ecological Environment of Urban and Rural Human Settlements under Grant GJJXK210105.

**Data Availability Statement:** Data on the distribution and proportion of arid and semi-arid regions in China are from the website: https://www.cma.gov.cn/kppd/kppdqxsj/kppdtqqh/201212/t20121215_197083.html (accessed on 11 March 2023).

**Conflicts of Interest:** The authors declare no conflict of interest.

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
