# Peer review of "Agroecological Risk Assessment Based on Coupling of Water and Land Resources—A Case of Heihe River Basin"

_land, doi:10.3390/land12040794_

Round 1

Reviewer 1 Report

This manuscript uses a land risk metric and water risk metric to assess ecological risk for a region in China, and then calculate the risk associated with coupling the two. Midstream political units appear to have the highest consistent risk.  

The abstract states there is spatial autocorrelation, but I find none in this paper.  

The overall English needs a review to clean up the grammar. It can be hard to read in places.

Overall the objective is a bit unclear-- to effectively evaluate risk? it doesn't really do that. To provide technical support? I don't see any technical applied results. And I don't think "Matching WLR" is the right term-- I would suggested "Risk-Correlated WLR". Risk appears to have been worse in specific areas mainly in the past-- however, the overall amount of coupling (the "matching coefficient") is converging to the 35%-50% range.    

Throughout, there are phrases I can't understand, like "matching coefficient of WLR is the driving factor" and "intensity of water connotation"? I don't know what these mean. The language is technically careful, but not precise, and reading is challenging because of word choice. I suggest a few terminology changes: Agro-ecological of water risks => water risk metric Agro-ecological risk of land => land risk metric Matching coefficients => matching suitability  

line 195: "with set to 0.5;" do you mean "with equal contribution set to 0.5"?  

Table 2: you call this the "Coupled agro-ecological risk coordination index", but on line 196 you call this the "coupled index". Be consistent.  

Tables 3-5. You report very high accuracy, with four significant digits. I don't see any evidence that you have this level of accuracy. Your data are based on inputs at millions of cubic meters. There is bound to be very high uncertainty in these measurements. Either check your significant digits, or report your uncertainty in these indexes in Tables 3-5.  

Figure 3: Agro-ecological risk of land is written twice in the figure. please fix.

I understand that the HRB data were only available in 5-year periods. However, it looks like there may exist multi-year conditions that caused stress-- e.g. 2000-2010 in the northern parts of Figure 3. Would you recommend more frequent data collection than 5-year intervals? Is risk more transient than you are able to detect at 5-year samples?  

Figure 5. "Matching coefficient agricultural of water and land resources" --- i think this should be "WLR matching coefficient".   Figure 5 should use a divergent color scale, not a color ramp. Also, Figure 5 does not tell me anything about how the government should allocate resources. In aggregate, it looks like Sunan County is OK, but the rest of the southern counties are not. In particular, Jiayuguan City is at the highest risk overall.   Is the coupling strength decreasing over time? why?  

The conclusions say that irrigation should be vigorously promoted, but the work does't really examine irrigation specifically -- it only uses aggregated risk indexes. If the matching coefficient is correlated with population, you should show this. 

Author Response

Dear Reviewer:

Dear Reviewer, Thank you very much for your detailed and informative review of this article, which resulted in many constructive comments and ideas. I have made changes to all the issues you have mentioned.

Additional explanatory diagrams and tables are included in the annex.

Question 1: The abstract states there is spatial autocorrelation, but I find none in this paper.

Answer 1 : This may have been a problem with my language, and I have corrected it in the abstract

Question 2: The overall English needs a review to clean up the grammar. It can be hard to read in places.

Answer 2: We apologize for the poor language of our manuscript. We worked on the manuscript for a long time and the repeated addition and removal of sentences and sections obviously led to poor readability. We have now worked on both language and readability and have also involved native English speakers for language corrections. We really hope that the flow and language level have been substantially improved.

Screenshot of some of the article retouching process

Question 3: Overall the objective is a bit unclear-- to effectively evaluate risk? it doesn't really do that. To provide technical support? I don't see any technical applied results. And I don't think "Matching WLR" is the right term-- I would suggested "Risk-Correlated WLR". Risk appears to have been worse in specific areas mainly in the past-- however, the overall amount of coupling (the "matching coefficient") is converging to the 35%-50% range.

Answer 3: Thank you for your suggestion. As agriculture is an important industry supporting production and livelihoods in arid zones, and the water and land ecology on which agriculture in arid zones depends is very fragile, it became the objective of this study to assess the agro-ecological risks of coupled agricultural water and land resources from the input-output perspective of water and land resources. The results of the assessment show that ecological risks differ between upstream, midstream and downstream, which allows for the identification of priority areas for future improvement work. The results show that "changes in the WLR matching coefficient in HRB are positively correlated with changes in the ecological risk of agricultural WLR use", resulting in high risk and high coupling in the past time period, which suggests that for arid zones, higher water-land coupling does not mean lower agricultural risk, but may be due to the special properties of arid zone crops or other factors such as groundwater. This has helped to screen strategic considerations for agroecological improvement. In addition, I have changed the term "matched WLR" to "risk-related WLR" in the text.

Question 4: Throughout, there are phrases I can't understand, like "matching coefficient of WLR is the driving factor" and "intensity of water connotation"? I don't know what these mean. The language is technically careful, but not precise, and reading is challenging because of word choice. I suggest a few terminology changes: Agro-ecological of water risks => water risk metric Agro-ecological risk of land => land risk metric Matching coefficients => matching suitability

Answer 4: Thank you for your suggestion, I have modified the terms you mentioned, as you requested. For those of you who don't understand “matching coefficient of WLR is the driving factor”,I changed it to “Matching suitability of WLR drives Agro-ecological risk”,replace“intensity of water connotation”to“water retention intensity”.

Question 5: line 195: "with set to 0.5;" do you mean "with equal contribution set to 0.5"? 

Answer 5: Thank you for your suggestion, I apologize for the omission of the explanation of some of the formula term letters in this section during the transcription of the article. Your reference to "with set to 0.5" has been changed to “both  and  are set to 0.5.”

Question 6: Table 2: you call this the "Coupled agro-ecological risk coordination index", but on line 196 you call this the "coupled index". Be consistent.

Answer 6: Thanks for your suggestion, I have fixed it.

Question 7 : Tables 3-5. You report very high accuracy, with four significant digits. I don't see any evidence that you have this level of accuracy. Your data are based on inputs at millions of cubic meters. There is bound to be very high uncertainty in these measurements. Either check your significant digits, or report your uncertainty in these indexes in Tables 3-5.

Answer 7: Thanks for your suggestion, the indices in Tables 3-5 are measured based on vector land use data, statistical yearbooks and other data, with an accuracy of up to 3 decimal places (in km2 ) for forest land, cropland and other areas summarized in ArcGIS. I have included Appendix 1 at the end of the article , which shows the measured data, to demonstrate the accuracy of the report.

Question 8 : Figure 3: Agro-ecological risk of land is written twice in the figure. please fix.

Answer 8: Thanks for your suggestion, I have fixed it.

Question 9 : I understand that the HRB data were only available in 5-year periods. However, it looks like there may exist multi-year conditions that caused stress-- e.g. 2000-2010 in the northern parts of Figure 3. Would you recommend more frequent data collection than 5-year intervals? Is risk more transient than you are able to detect at 5-year samples? 

Answer 9: Thanks for your suggestion, due to the limited nature of data collection we have chosen the current 5 year study interval, which spans from 1995 to 2020. The yearbook statistics for the earlier years were not perfect and we cannot guarantee the accuracy of data collection for more frequent study intervals. Of course your advice is very necessary! And if the relevant authorities can provide more frequent data, the results measured by the methods in this paper will be more convincing.

Question 10 : Figure 5. "Matching coefficient agricultural of water and land resources" --- i think this should be "WLR matching coefficient".   Figure 5 should use a divergent color scale, not a color ramp. Also, Figure 5 does not tell me anything about how the government should allocate resources. In aggregate, it looks like Sunan County is OK, but the rest of the southern counties are not. In particular, Jiayuguan City is at the highest risk overall.   Is the coupling strength decreasing over time? why?

Answer 10: Thanks for your suggestion, Figure 5 shows the WLR matching coefficients for different districts for each study year, and as only one index is expressed, a colour scale is used. The WLR matching coefficient for the most recent study year (2020) is shown as f in Figure 5, which shows that more districts and counties in the middle reaches of the Black River Basin have poor WLR matching, which can guide the government in allocating resources. Figure 5 shows that the gap in WLR matching between districts and counties decreases over time.

Question 11 : The conclusions say that irrigation should be vigorously promoted, but the work does't really examine irrigation specifically -- it only uses aggregated risk indexes. If the matching coefficient is correlated with population, you should show this. 

Answer 11: Thanks for your suggestion. The composite risk index in this paper is measured from an input-output perspective. The input indicators include agricultural water consumption and agricultural land area, which to some extent reflect the efficiency of irrigation, so the recommendation to promote irrigation is in line with the findings of the study.

Reviewer 2 Report

Thanks to the editor for the invitation. I think this is an interesting study and its research method has some practical significance and academic value. However, the article does not describe the applicability of the new method in enough detail, and there is a lack of consistency in the use of the method and in the discussion and conclusion sections. I think the article needs major revision, and my general judgment comes from the following specific questions and comments.

1. It is recommended that the literature review of studies on ecological risks of soil and water resource use be further revised and supplemented. For example, there is no connection between the " There is not much literature on ecological risk research using input-output methods" and the "lack of studies on soil and water resources coupling" in the previous section. In addition, the literature on the content and methodological system of ecological risk studies is less organized in this part, and it is suggested that it be added.

2. L197 The risk coupling coefficient of soil and water is mainly calculated by referring to the risk coefficient of transportation system.

3. L223 The authors chose to use the coupling index as the dependent variable, but the relationship between the matching coefficient of WLR and agricultural ecological risk changes is analyzed in chapters 3.2.3 and 5.2. Why the coupling index is used as the dependent variable instead of matching coefficient?

4. L379 The title of chapters 4.4 is not consistent with the following content. The title refers to ecological risks to land resources, while the text analyzes ecological risks to water resources. It is recommended that the analysis be carried out for the content of the title.

5. There are some minor errors in the text, for example, L49 should be changed from “water and water resources coupling” to “water and land resources coupling”. Please check the abbreviations carefully in the whole text.

Author Response

Dear Reviewer:

Dear Editor, Thank you very much for your detailed and informative review of this article, which resulted in many constructive comments and ideas. I have made changes to all the issues you have mentioned.

According to the suggestions you mentioned in question 1,I have added some summaries of ecological risk evaluation studies and pointed out the position of this article in ecological risk evaluation, and for "There is not much literature on ecological risk research using input-output methods There is not much literature on ecological risk research using input-output methods" "lack of studies on soil and water resources coupling" in the previous section" may be caused by my poor language expression, I have I have revised it in detail.

In response to your reference to question 2, many existing studies on land and social resource matching have made reference to the calculation of coupling coefficients for transport systems. For example, Zhang Wenzhong et al. applied coupling coefficients to study the relationship between land use change and industrialization and urbanization in the Pearl River Delta; Li Cunfang et al. applied coupling coefficients to study the system matching between resource-based enterprises in eastern China and resource-rich areas in the west.(ZHANG W Z, WANG C S, LV X, FAN J. Coupling relationship between land use change and industrialization & urbanization in the Zhujiang River Delta[J]. ACTA GEOGRAPHICA SINICA-CHINESE EDITION-, 2003, 58(5): 677-685; LI C F, WANG M L, ZHANG X X, DU S Y, ZHANG B. System Coupling Study between East Resources-based Enterprises and West Resource-rich Land[J]. Management Review, 32(10): 83.)

In response to your reference to question 3: As the ultimate aim of this study is the synergistic management of water and soil resources, rather than the management of a single water or soil resource, a coupling index is used to link the two together, representing coupled agro-ecological risks, with matching coefficients for driving analysis.

Based on your suggestion in question 4, I have revised the title of the article to "The Relationship Between Water Ecological Risk And Matching Coefficient" in order to make the title more consistent with the subsequent content analysis

Based on your suggestion in question 5, I have checked the full abbreviations in detail and made the changes you mentioned.

Reviewer 3 Report

I consider the article to be well-designed with an understandable and logical argumentation. The human element certainly plays a fundamental role in the observed phenomena, and the obtained results as well as the appropriately structured discussion confirm this fact.

I can only complain about the little things:

Neither the methodology nor the graphic outputs mention the data sources of the spatial layers boundaries and the software used to create the maps.

In line 247 or in Table 3, there is an incorrect number. I assume that the one that is part of the text is faulty.

Author Response

Dear Reviewer:

Dear Reviewer, Thank you very much for your detailed and informative review of this article, which resulted in many constructive comments and ideas. I have made changes to all the issues you have mentioned.

We have added information about the data sources and the software used to create the maps in the METHOD 3.1 section of the article. The Arcgis software was used for the output of the plots and maps in this study. We have also corrected the incorrect numbers you mentioned, it is indeed a problem in the text

Round 2

Reviewer 2 Report

The author has made the changes as per the comments and there are no more problems. Recommended for publication.